# Posture Estimation from Tactile Signals Using a Masked Forward Diffusion Model

**DOI:** 10.3390/s25164926

**Published:** 2025-08-09

**Authors:** Sanket Kachole, Bhagyashri Nayak, James Brouner, Ying Liu, Liucheng Guo, Dimitrios Makris

**Affiliations:** 1School of Computer Science and Mathematics, Kingston University, London KT1 2EE, UK; bhagyashrinayak26@gmail.com (B.N.); d.makris@kingston.ac.uk (D.M.); 2Department of Applied and Human Sciences, Kingston University, London KT1 2EE, UK; james.brouner@kingston.ac.uk; 3Tangi0 Ltd. (TG0), 73-75 Upper Richmond Road, London SW15 2SR, UK; ying@tg0.co.uk (Y.L.); liucheng@tg0.co.uk (L.G.)

**Keywords:** tactile pressure maps, posture estimation, convolutional-transformer neural network, diffusion models

## Abstract

Utilizing tactile sensors embedded in intelligent mats is an attractive non-intrusive approach for human motion analysis. Interpreting tactile pressure 2D maps for accurate posture estimation poses significant challenges, such as dealing with data sparsity, noise interference, and the complexity of mapping pressure signals. Our approach introduces a novel dual-diffusion signal enhancement (DDSE) architecture that leverages tactile pressure measurements from an intelligent pressure mat for precise prediction of 3D body joint positions, using a diffusion model to enhance pressure data quality and a convolutional-transformer neural network architecture for accurate pose estimation. Additionally, we collected the pressure-to-posture inference technology (PPIT) dataset that relates pressure signals organized as a 2D array to Motion Capture data, and our proposed method has been rigorously evaluated on it, demonstrating superior accuracy in comparison to state-of-the-art methods.

## 1. Introduction

Accurate estimation and understanding of human poses are essential in fitness applications, enabling real-time feedback, performance tracking, and personalized user experiences [1]. Traditionally, human pose estimation is solved using sensors or reflectors attached to the body or vision-based techniques. The former methods are intrusive and require time, effort, and special knowledge. The latter methods may be affected by occlusions, motion blur, etc., causing inaccuracies in the final outcome [2] and also require setting up the camera at an appropriate angle. Additionally, as the clamor for user privacy surges, the demand for non-vision-based systems intensifies, leading researchers to explore alternative modalities. Tactile-based 3D human pose estimation (HPE) aims to recover human 3D poses using tactile interactions between humans and the ground [3]. It has a wide range of potential applications such as augmented reality [4], robotics [5,6,7,8], sports analysis [9], the film industry [10], etc.

Recently, innovative pressure-tactile sensor arrays [11] have been developed to detect human movements [12] and recognize postures [13]. Although these studies have demonstrated the potential of using pressure images for pose estimation, they typically restrict their analyses to poses that involve significant body contact with the sensing surfaces [14]. In reality, high-quality tactile [15] information is often unavailable, especially since the obtained pressure maps frequently contain noise. Current methods tend to process these pressure maps manually or through hard-coded algorithms to estimate poses, limiting the scalability and versatility of pose estimation across diverse scenarios. Consequently, these approaches are confined to specific types of poses [3]. Moreover, the generated heat maps create blobs at human-ground contact points, making it challenging to distinguish between the feet of a single person or to differentiate between two individuals [16]. Achieving noise-free pressure maps is crucial for these systems to accurately predict human posture, particularly in applications like sports analytics, where minor variations in body movement can significantly affect the probability of winning [17]. Deriving 3D poses from sparse pressure imprints poses significant challenges for accurate motion capture, especially with minimal sensor contact, due to incomplete data and pressure map noise.

To date, the ability to extend tactile information from minimal contact areas to model detailed 3D human poses across a broad range of activities remains a formidable task. In addition, current methods [16] rely on synchronized tactile and visual frames to train models for human postures, facing limitations in precision due to the inherent ambiguity in interpreting visual data for complex poses, the computational and accuracy challenges of triangulating 3D keypoints from 2D detections, and the susceptibility to errors from occlusions and varying environmental conditions. These approaches are further constrained by the dependency on the initial accuracy of 2D keypoints extracted from RGB images, the computational intensity of the optimization processes, and the potential for over-smoothing in the application of 3D Gaussian filters, all of which can significantly affect the reliability and applicability of the pose estimation models.

In this paper, we introduce the improvement in tactile-based 3D human pose estimation. At the heart of our approach is a meticulously designed pressure mat embedded with a vast array of tactile sensors. These sensors capture real-time pressure signals when a person interacts with the mat, reflecting various postures and movements as shown in Figure 1. To tackle the shortcomings of visual frames in 3D keypoint ground truth generation, we propose a novel approach of using synchronized tactile signals and 3D keypoints as ground truth of critical body keypoints, offering a more accurate and direct mapping of human postures. While training requires both tactile carpet and body sensors, inference relies solely on the tactile sensor, making the system non-intrusive in practical use. In addition, this method leverages the high-resolution and unambiguous nature of motion capture data, circumventing the limitations of visual data interpretation and computational inefficiencies, thus enabling precise and scalable pose estimation across a wider range of activities and conditions. By employing a denoising diffusion model to generate noise-free pressure heatmaps, we predict the body’s 3D coordinates through a convolutional-transformer neural network, showcasing outstanding accuracy. Our approach offers a novel perspective on pose estimation, leveraging tactile information to overcome challenges posed by visual obstructions, thus presenting an unobtrusive and reliable method for interpreting human actions and interactions. The contributions of our work are as follows:A novel dual-diffusion signal enhancement (DDSE) architecture that adopts dual-forward diffusion processes. The noisy pressure signal and its associated morphological mask are each processed through their own forward diffusion pathways. At each step, features from both diffusion channels leverage a reverse diffusion process to denoise tactile information.A novel contour detection and alignment (CDA) layer, which integrates signals from dual-forward diffusion processes using spatial-pooling-based cross attention, significantly enhances spatial resolution by leveraging temporal information to enrich feature integration and refines contour detection from step-generated images.A pressure-to-posture inference technology (PPIT) dataset that combines tactile pressure maps with motion-captured data. This innovative motion-captured dataset addresses the challenges associated with image-based keypoint generation, thereby providing highly accurate ground truth for 3D keypoints.

The rest of this paper is organized as follows. Section 2 reviews related works. The proposed architecture is described in detail in the methodology Section 3. Section 4 provides experimental results and an ablation study. Finally, Section 5 presents the conclusion and scope for further research.

## 2. Related Work

### 2.1. Human Pose Estimation Using Tactile Sensing

Human pose estimation has advanced rapidly, with applications in interactive technologies, physical activity monitoring, augmented reality, gaming, sports analytics, and rehabilitation [18,19,20,21,22,23,24]. Traditional methods relied on probabilistic frameworks to analyze static images and infer relationships between body joints [25], while recent advancements have introduced deep learning techniques leveraging 3D supervisory signals, adversarial training, and multi-camera systems to address occlusions and ambiguities in 2D-to-3D inference [26,27,28]. Complementing these approaches, tactile sensing has emerged as a promising alternative, utilizing pressure-sensitive elements to capture complex pressure distribution patterns [29]. Tactile systems have been integrated into wearable devices like gloves and shoes [30] and non-wearable solutions such as smart beds and floors [31], demonstrating potential in activity recognition and motion analysis. Advanced techniques such as capacitive sensing [32], resistive and optical sensing, and piezoelectric materials have enabled tasks like walking pattern analysis, dynamic motion monitoring, and human localization [33]. Additionally, deep learning models, such as LeNet, combined with large-area fabric pressure sensor arrays, have successfully classified sitting postures with high accuracy [34], while pressure-sensing mats have been employed to infer 3D human pose and shape during rest [13]. However, despite these strides, tactile sensing systems have primarily focused on activity recognition and basic pose estimation, leaving the opportunity to innovate methods that integrate tactile data with alternative modalities, aiming to move beyond recognition toward accurate estimation of 3D human skeletons. Moreover, while hardware advancements have significantly improved tactile sensing, a gap remains in developing machine learning models tailored to high-resolution tactile datasets for accurate 3D human pose estimation.

### 2.2. Human Pose Estimation Systems

Advancements in motion capture for pose estimation have introduced varied techniques, each with its merits and limitations. Ref. [35] presents a single-view approach that uses exponential maps for tracking, but faces challenges with occlusions and complex movements. Ref. [36] employs Kinect for 3D estimation, which is limited by depth sensor resolution. Ref. [37] integrates physics for realistic motions from monocular videos, but demands high computational resources and precise conditions. Ref. [38] enhances motion capture with a balanced feedback mechanism, showing promise in controlled settings, but is limited in complex environments. Markerless capture methods, such as those by [39], use optical flow and multi-view sequences for detailed motion without physical markers, requiring extensive setup. Ref. [40] proposes a real-time algorithm using calibrated webcams that faces difficulties with occlusion, while Ref. [41] introduces a space-time shape approach that offers novelty but lacks generalizability to unpredictable movements. The prevailing gap in the literature is the lack of methodologies that leverage motion capture as a ground truth for estimating poses from tactile signals. Current strategies focus on visual and depth data, overlooking the potential of tactile information to provide a complementary and possibly more nuanced understanding of human movement.

### 2.3. Diffusion Models

Diffusion models, grounded in a probabilistic framework, iteratively transform noisy data into clean signals, making them particularly effective for denoising applications, including pressure signal analysis [42]. Among the neural backbones used within diffusion pipelines, the U-Net’s symmetric encoder–decoder architecture with skip connections that merge feature maps of equal resolution effectively preserves fine spatial detail while global context [43]. Accordingly, U-Net variants have become the default denoising core in many state-of-the-art diffusion frameworks, including those applied to pressure-signal reconstruction. DeepDeblur [44] and MPRNet [45] adopt convolutional architectures with distinct focuses, as follows: DeepDeblur uses multi-scale CNNs to refine image details at various scales, while MPRNet combines parallel feature extraction with multi-stage reconstruction for handling complex motion blur. In contrast, HINET [46] and Stripformer [47] introduce novel structures aimed at balancing computational efficiency and performance. HINET leverages a half-instance normalization block to maintain speed and accuracy, and Stripformer utilizes hybrid transformers to handle dynamic scenes by capturing strip-based tokens. Diffusion-based approaches, such as HI-Diff [48] and DID [49], focus on iterative refinement using hierarchical diffusion processes or learned noise distributions, whereas SI-DDPM-FMO [50] and Swintormer [51] enhance restoration through feature map optimization or adaptive attention mechanisms integrating convolutional and transformer-based models. While these methods exhibit diverse strategies, they share a common challenge capturing unwanted noise and struggling to denoise sparse input images due to architectural constraints. Their reliance on a single forward diffusion process often leads to misalignment between synthesized distributions and target results, underscoring the potential of dual-forward diffusion processes for improved accuracy and robustness.

### 2.4. Datasets for Tactile-Based HPE

Several datasets have been developed for tactile-based HPE, each providing unique insights into human posture recognition. Weibing et al. [34] introduced a dataset that captures sitting postures using a pressure sensor array on chairs, focusing on various common sitting positions to enhance posture recognition accuracy. Henry et al. [13] utilized a pressure mapping system to collect data on different standing and sitting postures, aiming to improve the classification of body positions in dynamic environments. Luo et al. [16] developed a dataset that integrates tactile signals from intelligent carpets to estimate 3D human poses, capturing various activities and providing a comprehensive view of posture dynamics. Chen et al. [52] focused on creating a dataset that combines tactile data with visual information to enhance the accuracy of posture estimation in diverse scenarios.

However, the currently available datasets often rely on camera-based systems, which raise privacy concerns and may not ensure the anonymity of individuals during data collection. The requirement for tactile pressure mats, paired with corresponding motion capture (MoCap) keypoint data, is crucial for achieving high accuracy in pose estimation, as it allows the integration of detailed physical interaction information with the precise spatial positioning of body parts.

## 3. Methodology

### 3.1. Pressure to Posture Estimation

Employing a carpet embedded with tactile sensors for monitoring human activities ensures privacy, which is lacking in camera-based systems. However, this method’s lower resolution compared to visual recordings presents challenges, notably the introduction of noise in the collected pressure data. In addition to reducing the sparsity in the single pressure map, the traditional approach of combining information from consecutive frames, used in [52], fails to capture the context of activities specific to a frame. The following limitation was empirically observed: distinctive actions confined to a subset of frames can be obscured or overshadowed by information in the later frames, inadvertently introducing unwanted noise. The pose in each frame may differ significantly, and concatenating consecutive frames risks blending these unique poses into an averaged representation, losing crucial temporal details. Consequently, such an approach risks missing crucial details, further compounding the noise issue with the inclusion of numerous frames.

The architecture detailed in Figure 2 for estimating posture from pressure maps involves a comprehensive two-stage process. Initially, in the first stage, as elaborated in Section 3.3, a sparse and noisy pressure signal collected from a pressure mat undergoes a forward and reverse denoising process in a diffusion model to generate a dense, noise-free pressure map, effectively eliminating the need to manually combine multiple pressure images for denoising. To further refine the learning process, two streams of the forward-noising process are employed. The second, masked stream incorporates a refined mask that serves as an attention mechanism, ensuring the model focuses on target pressure blobs. This additional stream mitigates the model’s tendency to learn noisy signals and helps avoid misinterpreting similar-looking noise as meaningful data, thereby enhancing the accuracy and robustness of the segmentation process. Refining noisy tactile signals represents significant progress over traditional tactile-based posture estimation techniques. Subsequently, in the second stage described in Section 3.4, the denoised tactile signal is fed into a transformer-convolution-based neural network, using 3D keypoints from MoCap data as labels for accurate human pose prediction.

### 3.2. Problem Definition

The primary objective of this study is to estimate the 3D posture keypoints $\hat{C}\left(t\right)$, which represent the human body pose at step *t*, from the sparse and noisy pressure signals P(t) acquired from a tactile mat. Mathematically,$$\hat{C}\left(t\right)=f\left(P\left(t\right)\right),$$

To address this, a two-stage framework is proposed:1.Stage 1: Denoise P(t) to reconstruct $P_{\mathrm{true}}\left(t\right)$, thereby reducing noise and enhancing the input signal.2.Stage 2: Use the denoised signal $P_{\mathrm{true}}\left(t\right)$ to estimate $\hat{C}\left(t\right)$ with high precision.

### 3.3. Stage 1: Dual-Diffusion Signal Enhancement (DDSE)

#### 3.3.1. Sparse/Noisy Pressure Signal

Let the pressure signal *P* be acquired directly from the tactile mat. This signal is often compromised by various forms of noise *N* due to environmental factors, sensor inaccuracies, or other disturbances. The aim of stage 1 of the methodology depicted in the upper part of Figure 2 is to extract the actual real pressure $P_{\mathrm{true}}$ applied by a person moving or standing on the mat.(1)$$P=P_{\mathrm{true}}+N$$

#### 3.3.2. Pressure Signal Forward Diffusion (PSFD)

The pressure signal undergoes a forward diffusion process over *D* steps, gradually adding noise. At each step *t*, the signal $P_{t-1}$ from the previous step is combined with Gaussian noise $\epsilon_{t}$. The factor $\alpha_{t}$, which is a time-dependent scalar, controls the proportion of the original signal and the noise in the current signal $P_{t}$. $\alpha_{t}$ is defined as a monotonically decreasing function of *t*, calculated using a linear decay schedule, ensuring that the influence of the original signal decreases while the noise contribution increases as the process progresses. This schedule was selected to ensure smooth transitions across steps and to balance the gradual addition of noise throughout the forward diffusion process.

Let $P_{t}$ be the pressure signal at step *t*, $\alpha_{t}$ be the factor controlling the noise level as previously discussed, and $\epsilon_{t}$ be the Gaussian noise added at each step, then mathematically, we have the following:(2)$$P_{t}=\sqrt{\alpha_{t}}P_{t-1}+\sqrt{1-\alpha_{t}}\epsilon_{t},\;\epsilon_{t}∼N\left(0,I\right),\;t=1,2,\ldots ,D$$

#### 3.3.3. Refined Mask Generation

In analyzing pressure signals, a critical step is creating a refined mask that accurately differentiates the foreground (areas with actual pressure) from the background (non-pressure areas). The initial mask is directly derived from the input tactile signal. As depicted in the upper part of Figure 2–Stage 1, a binary mask is generated by thresholding the input (t=0). Furthermore, it undergoes a morphological opening operation to remove noise. The opening operation, combining three successive erosions followed by three dilations with a 5 × 5 kernel, effectively eliminates small noise-related blobs within the pressure signal mask without significantly affecting the larger, significant pressure areas.

#### 3.3.4. Refined Soft Mask Forward Diffusion (RMFD)

Parallel to the pressure signal, the refined mask undergoes a forward diffusion process. Initially, the mask *M* is binary, but as Gaussian noise is incrementally added over *D* steps, it transitions into a soft mask. This process mirrors the transformation applied to the pressure signal, ensuring synchronization between the two during the reverse denoising stage. It ensures that the features and contours extracted from the soft mask can be effectively aligned and applied to the corresponding stages of the pressure signal. If $M_{t}$ is the mask at step *t* then mathematically, we have the following:(3)$$M_{t}=\sqrt{\alpha_{t}}M_{t-1}+\sqrt{1-\alpha_{t}}\epsilon_{t},\;\epsilon_{t}∼N\left(0,I\right),\;t=1,2,\ldots ,D$$

#### 3.3.5. Reverse Denoising Process

In our novel reverse denoising process, we implement a modified U-Net [43] model, which is uniquely modified with a key innovative contour detection and alignment (CDA) [53] layer as shown in Figure 3. This layer, composed of pyramidal pooling (PP) with 2, 4, 6, 8 atrous convolutions [54] and the cross-attention mechanism (CA) [55], is positioned at the U-Net model’s entrance. Its core function is to merge contour features from the masked signal into the tactile signal, thus enhancing the initial input for the U-Net. The significance of the CDA layer is to effectively integrate contour features from the masked signal into the corresponding noisy pressure signal, enhancing the model’s denoising efficacy.

As detailed in Section 3.3.2, the pressure signal $P_{t}$ and the mask $M_{t}$ undergo forward diffusion, accumulating noise incrementally over D steps. The noisy images resulting from this process at each step form the input to the CDA layer. This procedural approach allows the CDA layer to precisely estimate and integrate contour information relevant to the current noise state at each step. The CDA layer’s ability to focus on high-contour areas within the masked tactile image and align these features with the noisy pressure signal aids the U-Net model in more accurately predicting and extracting noise from the pressure signal. Simultaneously, it refines the pressure signal’s features based on the contour information from the mask, thereby enhancing the reconstruction of the original pressure signal for more accuracy. The mathematical representation of this denoising process is as follows:(4)$${\hat{P}}_{t-1}=U_{\theta }\left({\hat{P}}_{t},M_{t},t\right)$$
where ${\hat{P}}_{t}$ denotes the estimated pressure signal at step *t*, $U_{\theta }$ symbolizes the U-Net model equipped with parameters θ, and $M_{t}$ represents the mask at step *t*, enhanced by the CDA layer. This iterative process iterates through the steps, progressively refining the signal quality at each stage.

The output of the reverse denoising is $P_{\mathrm{true}}$, the denoised pressure signal. This streamlined approach, combining the functionalities of the CDA layer and the U-Net model, not only diminishes noise but also sharpens and defines tactile contours, leading to an improved reconstruction of the pressure signal with each successive iteration.

### 3.4. Stage 2: 3D Pose Prediction Transformer (3DPPT)

#### 3.4.1. Transformer Encoder

The denoised tactile signal $P_{\mathrm{true}}$ is subsequently passed through a transformer encoder. The encoder consists of layers of layer normalization (LN) and self-attention (SA) mechanisms, which refine the features for precise posture estimation. The mathematical operation within the transformer encoder can be expressed as follows:(5)$$T\left(t\right)=\mathrm{TransformerEncoder}\left(\mathrm{LN}(\mathrm{SA}\left(P_{\mathrm{true}}\right))\right)$$
where T(t) represents the encoded feature set prepared for the decoding stage.

#### 3.4.2. Decoder Stage

Finally, the decoder stage comprises a series of deconvolution (Deconv), batch normalization (BN), and rectified linear unit (ReLU) layers, which upsample and normalize the features before making the final keypoint predictions. The decoder operation is mathematically described by the following:(6)$$K\left(t\right)=\mathrm{ReLU}\left(\mathrm{BN}\left(\mathrm{Deconv}(T(t))\right)\right)$$
The keypoint prediction layer maps the processed features from the decoder to 3D pose keypoints during training. This regression step transforms the feature map into specific keypoint predictions, which are represented as follows:(7)$$\hat{C}\left(t\right)=\mathrm{KeypointPredictor}\left(K\left(t\right)\right)$$
where $\hat{C}\left(t\right)$ represents the predicted pose keypoints at time *t*, and K(t) is the feature map from the decoder. The decoder output is first flattened into a 1D vector to ensure compatibility with the fully connected layer. The fully connected layer applies a linear transformation to map the flattened tactile image features to the 3D pose keypoints:(8)$$\hat{C}\left(t\right)=WK_{\mathrm{flat}}\left(t\right)+b$$

Here, *W* is the weight matrix that defines the mapping from the high-dimensional features to the keypoints, and *b* is the bias vector added to the output. The final output $\hat{C}\left(t\right)$ is a 12-dimensional vector representing the predicted 3D pose keypoints.

## 4. Experimental Evaluation

### 4.1. Dataset

We introduce the novel PPIT dataset to assess the effectiveness of the proposed method. The PPIT dataset is an extensive collection of synchronized tactile signal frames and 3D pose keypoints, designed to enable human pose estimation through foot pressure measurements. Tactile signal frames (resolution 496 × 298) are captured using TG0 Advanced Pressure Mats [56], designed and manufactured by TG0. Each 60 cm × 30 cm module implements an approximately 15 × 8 grid of capacitive tactels set on a 40 mm pitch lattice. Two identical sensing layers are stacked; the upper layer records binary contact-area information, while the lower layer measures normal pressure, providing 240 raw channels (120 area + 120 pressure) per module. Tiling the six modules in a 3 × 2 arrangement yields an effective sensing area of 1.80 m × 0.60 m (45 × 16 tactels). The pressure layer resolves forces up to 15 kPa with a sensitivity of 0.1 kPa. The TG0 Advanced Pressure Mat is designed for real-time applications, streaming data at 60 Hz with onboard calibration and a low-latency API. The capacitive sensing surface demonstrates negligible residual deformation and recovers within a few milliseconds under normal body-weight loads. The mat reliably captures quasi-static and moderately dynamic movements (e.g., walking, posture transitions), while very fast ballistic motions may require higher sampling rates or complementary sensors.

The ground truth, in the form of 3D pose keypoints, is obtained through a motion capture system comprising nine high-resolution Arqus-700 cameras [57], which track three markers attached to each of 12 key body points. Each camera features a 26 MP sensor capable of capturing detailed motion at 200 Hz with an impressive 3D resolution of 0.3 mm.

The dataset consists of 12 distinct activities, each performed by a volunteer. Activities range from various postures like squatting and goddess yoga positions to dynamic movements such as forearm plank transitions and push-ups; see Table 1. Each action was performed for approximately one minute. The dataset contains 60,000 tactile signal frames. We separated two activities (seating and squatting) from the 12 to serve as the validation dataset. The model was trained on the remaining 10 poses, which were further split into 80% for training and 20% for testing. The recorded tactile frames and pose data were synchronized based on their timestamps, ensuring that the tactile frame P(t) corresponds to the 3D pose keypoints K(t) at the same moment *t*. Our dataset is the first to offer tactile signals paired with accurate motion capture data. A wide range of activities, the corresponding pressure signals, and the motion-captured skeleton are shown in Table 1.

### 4.2. Experimental Protocol

The Stage 1 dual-forward diffusion and Stage 2 3D-pose-prediction transformer models were trained using mean squared error (MSE) loss and optimized with the Adam optimizer. They were trained on a The GPU PC system features 128 GB of memory, an Intel Xeon W-2155 processor (Intel Corporation; Santa Clara, CA, USA), and NVIDIA Quadro RTX 8000 48 GB graphics cards (NVIDIA Corporation; Santa Clara, CA, USA), housed in a Lenovo ThinkStation P520 chassis (Lenovo Group Ltd.; Beijing, China). No other similar datasets with paired 3D poses are currently available, so validation was performed only on the PPIT dataset. We conducted experimental evaluations of the two stages separately. Stage 1, evaluation of pressure signal restoration (Section 4.5), assesses the denoising and signal restoration capabilities of the model. Stage 2, posture prediction evaluation (Section 4.6), compares the model’s ability to predict 3D poses.

### 4.3. Stage 1 Evaluation Metrics

#### 4.3.1. Peak Signal-to-Noise Ratio (PSNR)

The peak signal-to-noise ratio (PSNR) metric is used for measuring the quality of a reconstructed or processed image compared to its original (reference) version. PSNR, measured in decibels (dB), is based on the error between corresponding pixels in the two images and is quantified using the mean squared error (MSE). The ratio essentially compares the image’s maximum possible pixel value (peak signal) to the power of its noise (distortion or error). High PSNR values indicate lower distortion and, thus, higher image quality.(9)$$\mathrm{PSNR}=10\cdot \log_{10}\left(\frac{{\mathrm{MAX}}_{I}^{2}}{\mathrm{MSE}}\right)$$
where ${\mathrm{MAX}}_{I}$ is the maximum possible pixel value of the image (e.g., 255 for 8-bit images) and MSE is the mean squared error between the original and the processed image.

#### 4.3.2. Structural Similarity (SSIM)

Structural similarity (SSIM) is a metric used for measuring the similarity between two images. Unlike PSNR, which primarily focuses on pixel-wise errors, SSIM considers changes in structural information, luminance, and contrast. The SSIM index aims to provide a more perceptually relevant measure by accounting for the fact that the human visual system is highly adapted for extracting structural information from a visual scene. A higher SSIM value (closer to 1) implies greater similarity between the compared images. SSIM is a unitless metric ranging from 0 to 1, where a higher value (closer to 1) implies greater similarity between the compared images.(10)$$\mathrm{SSIM}\left(x,y\right)=\frac{\left(2\mu_{x}\mu_{y}+c_{1}\right)\left(2\sigma_{xy}+c_{2}\right)}{\left(\mu_{x}^{2}+\mu_{y}^{2}+c_{1}\right)\left(\sigma_{x}^{2}+\sigma_{y}^{2}+c_{2}\right)}$$
where *x* and *y* are the two images being compared, $\mu_{x},\mu_{y}$ are their average pixel values, $\sigma_{x},\sigma_{y}$ are their variances, $\sigma_{xy}$ is the covariance, and $c_{1},c_{2}$ are constants to stabilize the division.

#### 4.3.3. Learned Perceptual Image Patch Similarity (LPIPS)

Learned perceptual image patch similarity (LPIPS) is a perceptual metric that evaluates image similarity by computing the distance between feature representations extracted from a pre-trained neural network, providing a measure aligned with human visual perception. LPIPS is a unitless metric ranging from 0 to 1, where lower values indicate greater perceptual similarity. (11)$$\mathrm{LPIPS}\left(x,y\right)=\sum_{l}w_{l}{∥\phi_{l}\left(x\right)-\phi_{l}\left(y\right)∥}_{2}^{2}$$
where *x* and *y* are the two images being compared, $\phi_{l}\left(x\right)$ and $\phi_{l}\left(y\right)$ represent the feature maps from layer *l* of the network for images *x* and *y*, $w_{l}$ is the weight for layer *l*, and ${∥\cdot ∥}_{2}$ is the Euclidean norm.

### 4.4. Stage 2 Evaluation Metrics

#### 4.4.1. Mean per Joint Position Error (MPJPE)

Mean per joint position error (MPJPE) measures the average Euclidean distance, in millimeters (mm), between the predicted and ground truth positions of various joints in the human body. It is a key indicator of the accuracy of a pose estimation model, with a lower MPJPE value indicating higher accuracy.(12)$$\mathrm{MPJPE}=\frac{1}{N}\sum_{i=1}^{N}\sqrt{\sum_{j=1}^{J}{∥P_{ij}-G_{ij}∥}^{2}}$$
where *N* is the number of samples, *J* is the number of joints, $P_{ij}$ is the predicted position of the *j*th joint in the *i*th sample, and $G_{ij}$ is the corresponding ground truth position.

#### 4.4.2. Average Keypoint Localization Error of Whole Body (AKLEB)

The average keypoint localization error of whole body (AKLEB), measured in millimeters (mm), offers a holistic assessment of whole-body pose estimation accuracy, contrasting with MPJPE’s focus on joint localization. AKLEB evaluates the localization accuracy of all key body parts, providing a comprehensive measure of a model’s ability to capture the body’s pose nuances. A lower AKLEB indicates a more accurate pose estimation across the entire body, which is crucial for detailed and precise body movement analysis.(13)$${\mathrm{AKLEB}}_{d}=\frac{1}{N}\sum_{i=1}^{N}\frac{1}{J}\sum_{j=1}^{J}∥P_{ijd}-G_{ijd}∥$$
where *N* is the number of samples, *J* is the total number of joints across the whole body, $P_{ijd}$ is the predicted coordinate for dimension d∈[X,Y,Z] of the *j*th joint in the *i*th sample, and $G_{ijd}$ is the corresponding ground truth.

### 4.5. Evaluation of Pressure Signal Restoration

The efficacy of image restoration methods is differentiated through a meticulous examination of performance metrics, including PSNR, SSIM, mean average error (MAE), and LPIPS, applied to the tactile frames of the tactile signal frames of the PPIT dataset. The proposed methodology demonstrates superior restoration capability against state-of-the-art methods as evidenced by the quantitative metrics tabulated in Table 2. Among the evaluated methods, diffusion models like SI-DDPM-FMO and Swintormer lead in performance, with our method surpassing all others in PSNR (36.24) and SSIM (0.873) while demonstrating the lowest MAE (0.045) and LPIPS (0.109). SI-DDPM-FMO struggles with motion blur overlapping background elements, while Swintormer faces challenges with varied blurring scenarios due to a small compression ratio in the latent space, and DID might have inconsistencies in exposure and white balancing. Our method overcomes these limitations by leveraging dual-forward diffusion processes for enhanced noise and blur handling, along with the CDA layer for improved contour detection and alignment, ensuring better adaptation to complex scenarios and delivering superior image restoration outcomes.

#### Computational
Efficiency Comparison

In our image restoration technique comparison as shown in Table 3, our method not only achieves the highest PSNR of 36.24 dB but also demonstrates an optimal balance between computational efficiency and image quality restoration. The parameter count, indicating the trainable parameters within the architecture, stands at 135.41 million, while the multiply-accumulate operations (MACs) reflect our model’s complexity. This is comparable to DID and Swintormer, but they do not match our method’s PSNR, underscoring its superior balance of computational demand and restoration quality. Specifying trainable parameters alongside MACs highlights our model’s efficient design in achieving state-of-the-art image restoration.

### 4.6. Posture Prediction Evaluation

To conduct a fair comparison, we use the PPIT dataset to evaluate the posture prediction methods; their validation results are shown in Table 4. This approach is crucial given the inherent differences in the architecture and intended applications of these methods, as well as the variation in the datasets originally applied. In adapting methods such as [34], tactile input signals are mapped to posture classifications using fully convolutional networks (FCNs). We replaced the classification layer with a regression head. The transformer encoder in [52], which originally encoded temporal features for SMPL [59] parameter prediction, was modified to output keypoint embeddings. Additionally, the loss function, which used SMPL loss and reconstruction loss, was replaced with MSE for keypoint regression.

The results in Table 4 demonstrate the superior performance of our method compared to existing approaches. Our method achieves the lowest mean per joint position error (MPJPE) of 48.41 mm, outperforming Luo et al. [16], the next best performer, by a significant margin of 13.41 mm. Similarly, in terms of average keypoint localization error of whole body (AKLEB) coordinates, our method consistently produces the lowest errors across all dimensions (*X*, *Y*, and *Z*), highlighting its robustness in predicting 3D poses. Our method is the only tactile-based approach that produces comparable results to modern image-based pose estimation methods [60], despite the fundamentally different sensing modality.

The methods by [16,52] excel in 3D human pose estimation from tactile signals through adversarial learning, attention, and CNNs, navigating the challenges of noisy data. Our technique outperforms these by achieving the lowest MPJPE and AKLEB in all dimensions, showcasing superior accuracy in pose estimation. The key distinction of our method lies in its advanced handling of the intrinsic limitations posed by tactile pressure sensors, which produce noisy and sparse data. Unlike traditional approaches that rely on stitching consecutive pressure images to compile information, sacrificing temporal resolution in the process and adding unnecessary noise, our dual-diffusion and the CDA layer improve the quality of the input data. Additionally, while these methods generate ground truth through multiple processes, such as triangulation from 2D to 3D keypoints, which reduces the accuracy of the ground truth, we capture ground truth using motion capture, avoiding any processing and providing accurate 3D keypoints. Furthermore, we implement pyramidal pooling before utilizing a transformer-convolution architecture for keypoint prediction. This innovative strategy significantly boosts our method’s performance, enabling it to outshine conventional tactile-based methods in accuracy and efficiency.

### 4.7. Ablation Study

An ablation study was conducted to assess the impact of various novel architectural elements on mean per joint position error (MPJPE) illustrated in Table 5. The best-performing combination included PSFD, RMFD, PP, and CA elements, which resulted in the lowest MPJPE of 4.4382. Passing the features from PSFD and RMFD directly to CA improves accuracy; however, refining the features through atrous convolutions in PP yields the lowest MPJPE.

## 5. Conclusions

In this paper, we made significant strides in advancing tactile-based 3D human pose estimation systems that leverage tactile information through a high-density tactile carpet and corresponding motion-captured poses as ground truth. Our dual-diffusion signal enhancement (DDSE) model is utilized to restore the tactile signal in the pressure images, representing the pressure information at a higher temporal and spatial resolution. Furthermore, it uses a transformer-convolution architecture for posture prediction, outperforming state-of-the-art methods. A key contribution is our unique PPIT dataset, which combines tactile pressure maps with motion-captured data and is the first to address the challenges of image-based keypoint generation with highly accurate ground truth. Our approach has been rigorously validated on the PPIT dataset, showcasing its exceptional ability to capture the nuances of human movement with high accuracy and reliability. Our method’s validation showcases its superiority in key metrics, promising significant advances in fields like human–computer interaction and assistive technologies. In future work, we will address the issue of users with musculoskeletal asymmetries (e.g., pelvic tilt, flat feet, scoliosis, injuries) by extending our dataset with a wider range of participants, and sophisticated data augmentation methods [61,62,63]. By transforming tactile data into accurate pose predictions, our research introduces a groundbreaking approach to human pose estimation. This reliable, non-intrusive method surpasses conventional visual systems, establishing a benchmark for future tactile-based 3D human pose estimation.

## Figures and Tables

**Figure 1 sensors-25-04926-f001:**
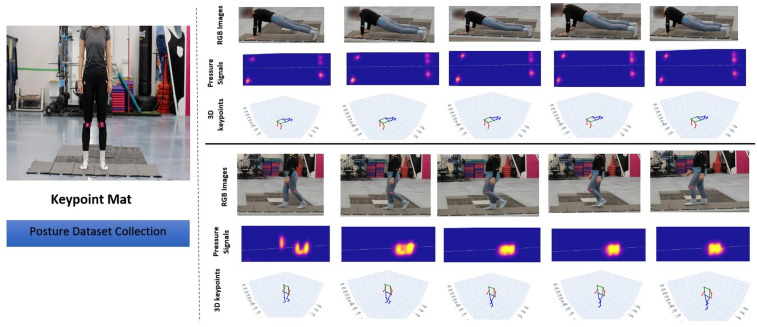
The figure illustrates the ‘pressure to posture’ technology, which utilizes tactile pressure data to predict body posture. It depicts a subject adorned with motion capture markers performing various exercises on a pressure-sensitive mat, facilitating the collection of posture data. The first row captures the subject in a plank position, while the second row displays corresponding pressure distribution maps from the tactile mat. The third row shows posture predictions made by a specially designed neural network. This process is repeated for the walking posture, as shown in the last three rows.

**Figure 2 sensors-25-04926-f002:**
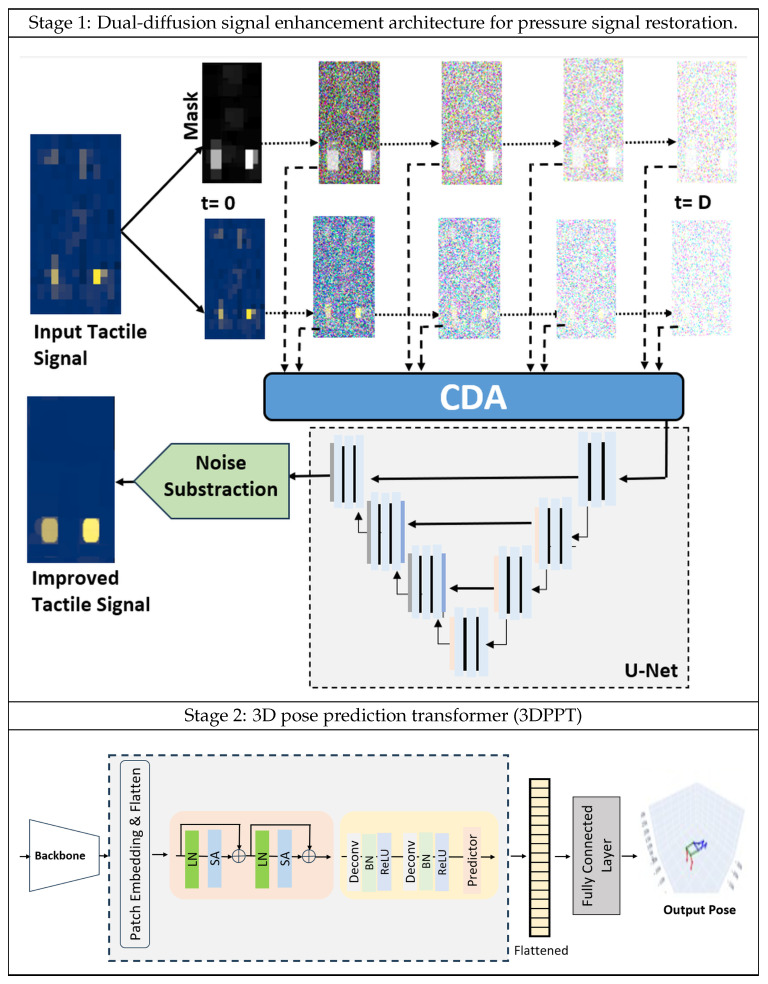
Proposed posture estimation framework with two stages. Stage 1 utilizes dual-forward diffusion for the noisy pressure signal and morphological mask, integrating features at each step via the CDA layer and performing denoising through reverse diffusion. Stage 2 employs a transformer-convolution neural network for 3D keypoint estimation.

**Figure 3 sensors-25-04926-f003:**
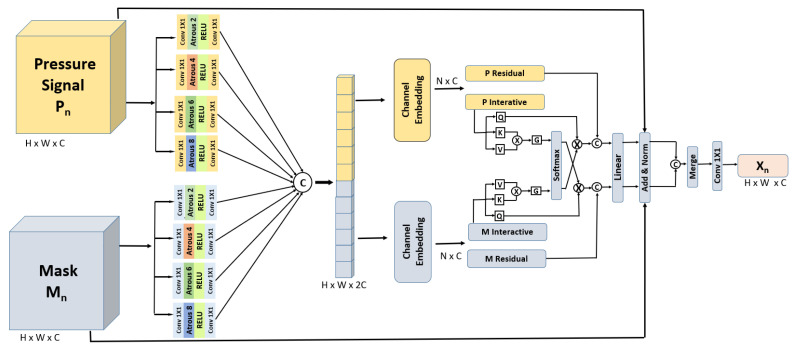
Contour detection and alignment (CDA) layer.

**Table 1 sensors-25-04926-t001:** Visualization of the PPIT dataset.

Pose Title	Pose	Pressure Map	Motion Captured Skeleton
Squat	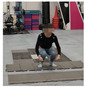	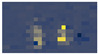	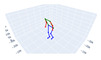
Stay in Goddess pose	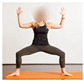	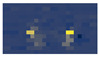	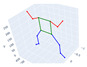
Extend legs	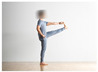	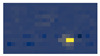	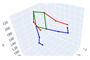
Stay Standing	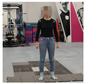	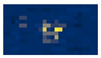	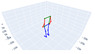
Bend upper body	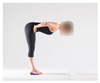	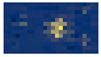	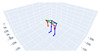
Standing wide-legged and forward fold	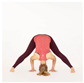	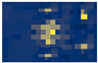	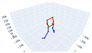
Plank	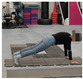	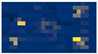	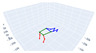
Right Lunge	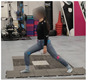	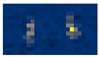	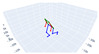
Walking	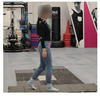	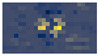	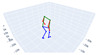
Sit	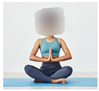	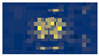	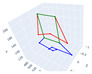
Sit Up	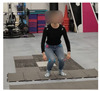	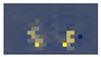	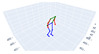
Left Lunge	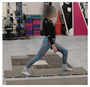	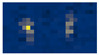	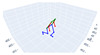

**Table 2 sensors-25-04926-t002:** Comparison of state-of-the-art image restoration methods on the PPIT dataset.

Method	PSNR (db) ↑	SSIM ↑	MAE (mm) ↓	LPIPS ↓
DeblurGAN [58]	23.95	0.614	0.057	0.315
DeepDeblur [44]	24.06	0.621	0.055	0.347
MPRNet [45]	26.48	0.758	0.054	0.348
HINET [46]	29.61	0.745	0.053	0.231
Stripformer [47]	30.34	0.734	0.052	0.214
Hi Diff [48]	34.71	0.714	0.05	0.271
DID [49]	35.5	0.842	0.051	0.201
SI-DDPM-FMO [50]	35.66	0.862	0.048	0.116
Swintormer [51]	35.68	0.821	0.049	0.013
**Ours**	**36.24**	**0.873**	**0.045**	**0.109**

*Notes:* **Bold** denotes the best value in each column; underline denotes the second-best; the green-shaded row highlights our method. ↑ means higher is better; ↓ means lower is better. PSNR is reported in dB; MAE in mm; SSIM and LPIPS are unitless.

**Table 3 sensors-25-04926-t003:** The multiply-accumulate operations are estimated when the input is 256×256. Our method outperforms existing baselines, achieving state-of-the-art quality while being computationally efficient.

Method	Param (M)	MACs (G)	PSNR (dB) ↑
Stripformer [47]	**36.13**	18.7	30.34
Hi Diff [48]	85.17	130.35	34.71
DID [49]	128.23	**6.52**	35.5
SI-DDPM-FMO [50]	131.53	15.43	35.68
Swintormer [51]	154.89	8.02	35.66
Ours	135.41	7.05	**36.24**

*Notes:* **Bold** denotes the best value in each column; underline denotes the second-best; the green-shaded row highlights our method.

**Table 4 sensors-25-04926-t004:** Pose prediction evaluation.

Method	MPJPE (mm)	AKLEB (mm)		
		X	Y	Z
Weibing et al. [34]	78.25	92.71	83.52	81.23
Luo et al. [16]	61.82	81.52	68.38	61.79
Wenqiang et al. [52]	65.21	74.62	65.15	59.65
Ours (Stage 1+ 2)	**48.41**	**73.75**	**63.8**	**56.93**

*Notes:* **Bold** denotes the best value in each column; underline denotes the second-best; the green-shaded row highlights our method.

**Table 5 sensors-25-04926-t005:** Evaluation of novel elements in architecture.

Novel Elements of Architecture				MPJPE
PSFD				16.7904
PSFD	RMFD	CA	-	6.7812
**PSFD**	**RMFD**	**CA**	**PP**	4.4382

*Notes:* **Bold** The green-shaded row highlights best performing combination of elements the proposed architecture.

## Data Availability

The original data presented in the study are openly available at https://github.com/tg0uk/PPIT_database, accessed on 5 November 2023.

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
