# Peer review of "Posture Estimation from Tactile Signals Using a Masked Forward Diffusion Model"

_sensors, 2025, doi:10.3390/s25164926_

Round 1
Reviewer 1 Report
Comments and Suggestions for Authors
The article is devoted to the estimation of 3D human pose based on tactile data. The methodology is described in detail and in a structured manner:
1. Due to the limited data for verification (there is only one dataset with tactile and 3D images of the object). How well will the pose of people with pelvic asymmetry, flat feet, etc., whose weight is not evenly distributed, be estimated?
2. How realistic is it to make such a real-time system? What is the minimum time required to restore the shape of the mat material? Will it be possible to perceive some very dynamic movements?
3. I would like to see scenarios with different types of noise (for example, systematic sensor errors).
4. In Table 3, you compare many methods on the selected dataset, although half of the studies selected for comparison were specialized in slightly different tasks, so it is logical that for your particular task they may not be the best. Is it possible to compare with works devoted to determining the pose from an image?
The paper requires further research to analyze generalizability, computational efficiency and applicability in real-world scenarios and can be published after revision.
Author Response
Reviewer 1 Addressed Comments:
- Due to the limited data for verification (there is only one dataset with tactile and 3D images of the object). How well will the pose of people with pelvic asymmetry, flat feet, etc., whose weight is not evenly distributed, be estimated?
Author reply –
Thank you for noting this gap. Our dataset of 12 healthy participants does not cover pelvic asymmetry, flat feet, or similar load-imbalancing conditions, so model performance in those cases remains untested. We now flag this as a future scope in the conclusion section at (p. 15, L. 464–467).
“In future work, we will address the issue of users with musculoskeletal asymmetries (e.g., pelvic tilt, flat feet, scoliosis, injuries), by extending our dataset with a wider range of participants, and sophisticated data augmentation methods [51-53]”.
- How realistic is it to make such a real-time system? What is the minimum time required to restore the shape of the mat material? Will it be possible to perceive some very dynamic movements?
Author reply –
We now provide further information regarding the specifications of the mat (P9, L306-311).
“The TG0 Advanced Pressure Mat is designed for real-time applications, streaming data at 60 Hz with onboard calibration and low-latency APIs. The capacitive sensing surface demonstrates negligible residual deformation and recovers within a few milliseconds under normal bodyweight loads. The mat reliably captures quasi-static and moderately dynamic movements (e.g., walking, posture transitions), while very fast ballistic motions may require higher sampling rates or complementary sensors.”
- I would like to see scenarios with different types of noise (for example, systematic sensor errors).
Author reply –
Thank you for the suggestion. This study does not explicitly model or evaluate different types of sensor noise (e.g., systematic errors). Instead, we apply morphological preprocessing techniques (erosion and dilation) to suppress background and transient noise in the pressure maps, as discussed in the paper (P7, L233-235). Our primary focus is on designing and validating a pose-estimation pipeline based on clean, preprocessed tactile input and dealing with the noise already present on the specific sensor. We agree that a more detailed analysis of sensor noise characteristics and their impact on prediction accuracy is a valuable direction for future work.
- In Table 3, you compare many methods on the selected dataset, although half of the studies selected for comparison were specialized in slightly different tasks, so it is logical that for your particular task they may not be the best. Is it possible to compare with works devoted to determining the pose from an image?
Author reply –
Image-based pose-estimation methods rely on a fundamentally different sensing modality and inference pipeline than pressure-mat approaches. To ensure a fair, like-for-like comparison, Table 3 and Table 4 includes only studies that estimate body pose from tactile or pressure data. We have now added the following sentence discussing how our method compares to image-based approaches (P14, L427-429):
“Our method is the only tactile-based approach that produces comparable results to modern image-based pose estimation methods [54], despite the fundamentally different sensing modality”

Reviewer 2 Report
Comments and Suggestions for Authors
Dear authors
I have overall enjoyed article reading. The topic discussed by the authors is interesting to the readers, and in general, the article is very well written. I list below some major and minor and changes that must be addressed before further article processing.
Major changes
Lines 170 – 176: Has this observation been stated elsewhere? If so, please cite.
Section 3.3.3: I could not understand how you set mask dimension (measured in pixels or sensing tactless). Does the mask change dynamically depending upon the force profile? Please specify.
Section 3.3.5 (Lines 236 – 241): This section discussed about some concepts that may not be fully understood by all readers such as U-net model and Contour Detection and Alignment (CDA) layer. Please consider defining them in Section 2 or in a separate appendix.
Minor changes
Line 9: What does PPIT stand for? Please make sure to define each acronym before using it.
Line 84: The introduction section provided a comprehensive overview of current challenges within the field of posture estimation using tactile sensors. However, it is recommended to include a paragraph before section 2 that discuses about the remainder of the paper, e.g. this paper is organized as follows…
Line 290: Please specify the model and manufacturer of the specialized mat. Is the mat commercially available or is it self-produced? Please specify. Regarding the number of sensing tactels within the mat, what are the physical dimensions of each tactel?
Tables 1 and 2 have the same title and this is somewhat confusing. Please consider using different titles for each table or simply merge them.
Author Response
Reviewer 2 Addressed Comments:
Major Changes
- Lines 170 – 176: Has this observation been stated elsewhere? If so, please cite.
Author reply –
The observation in Lines 179–185 stems from our own pilot experiments on the PPIT dataset. When we manually concatenated consecutive pressure frames, the resulting images exhibited noticeable blurring. To the best of our knowledge, this specific artefact has not been documented in the literature, so no external citation is available. However, we have mentioned that these are empirical observations.
- Section 3.3.3: I could not understand how you set mask dimension (measured in pixels or sensing tactless). Does the mask change dynamically depending upon the force profile? Please specify.
Author reply –
The binary mask is created in pixel space by applying three successive erosions followed by three dilations with a 5 × 5 kernel. This three-iteration setting was chosen empirically to suppress isolated noise while preserving pressure contours and remains fixed for all frames; the mask therefore does not adapt dynamically to the force profile. It is mentioned from Lines 228-235.
- Section 3.3.5 (Lines 236 – 241): This section discussed about some concepts that may not be fully understood by all readers such as U-net model and Contour Detection and Alignment (CDA) layer. Please consider defining them in Section 2 or in a separate appendix.
Author reply –
Thank you for the valuable comment. We have added the U-net significance in the section 2.3 at Lines 134-138.
The Contour Detection and Alignment (CDA) layer is a novel computational element introduced in this work; to the best of our knowledge, no prior publication describes an equivalent module, so an external citation is not available. We have now added citations in Section 3.3.5 for the established components it builds upon, namely atrous spatial-pyramid pooling \cite{chen2018deeplab}, U-Net \cite{ronneberger2015}, and the cross-attention mechanism \cite{vaswani2017}. Which is also briefly described in literature.
Minor changes
- Line 9: What does PPIT stand for? Please make sure to define each acronym before using it.
Author reply –
We have expanded the acronym on first use (line 9) to “Pressure-to-Posture Inference Technology (PPIT)” and use the abbreviation only after this definition.
- Line 84: The introduction section provided a comprehensive overview of current challenges within the field of posture estimation using tactile sensors. However, it is recommended to include a paragraph before section 2 that discuses about the remainder of the paper, e.g. this paper is organized as follows…
Author reply –
Thank you for the valuable comment. We have added a paragraph mentioning structure of the rest of the paper. Please refer the added paragraph on the lines 86-89.
- Line 290: Please specify the model and manufacturer of the specialized mat. Is the mat commercially available or is it self-produced? Please specify. Regarding the number of sensing tactels within the mat, what are the physical dimensions of each tactel?
Author reply –
The mat is commercially available. We have addressed the specification and other details of the mat in the paper on line 296-311:
- Tables 1 and 2 have the same title and this is somewhat confusing. Please consider using different titles for each table or simply merge them.
Author reply –
Thank you for noticing the minor errors. We have merged the two tables into one eg. table 1

Round 2
Reviewer 2 Report
Comments and Suggestions for Authors
Mandatory changes have been replied one by one. The article can be published in present form